# Subcutaneous ketamine infusion in palliative patients for major depressive disorder (SKIPMDD)—Phase II single-arm open-label feasibility study

**Wei Lee**[1,2,3,4]*, **Caitlin Sheehan**[5], **Richard Chye**[1,2,6,7], **Sungwon Chang**[1], **Adam Bayes**[7,8], **Colleen Loo**[7,8], **Brian Draper**[7], **Meera R. Agar**[1,7], **David C. Currow**[9]

1 University of Technology Sydney, Ultimo, NSW, Australia, 2 St. Vincent Health Australia, Sydney, NSW, Australia, 3 HammondCare, Royal North Shore Hospital, St. Leonards, NSW, Australia, 4 University of Sydney, Northern Clinical School, St. Leonards, NSW, Australia, 5 Calvary Hospital, Kogarah, NSW, Australia, 6 University of Notre Dame Australia, Fremantle, NSW, Australia, 7 University of New South Wales, Randwick, NSW, Australia, 8 Blackdog Institute, Hospital Road, Prince of Wales Hospital, Randwick, NSW, Australia, 9 University of Wollongong, Wollongong, NSW, Australia

* Wei.lee@health.nsw.gov.au

**Data Availability Statement:** There are no competing interests that affect our adherence to

## Abstract

### Background

Ketamine at subanaesthetic dosages ($\leq$0.5mg/kg) exhibits rapid onset (over hours to days) antidepressant effects against major depressive disorder in people who are otherwise well. However, its safety, tolerability and efficacy are not known for major depressive disorder in people with advanced life-limiting illnesses.

### Objective

To determine the feasibility, safety, tolerability, acceptability and any antidepressant signal/ activity to justify and inform a fully powered study of subcutaneous ketamine infusions for major depressive disorder in the palliative setting.

### Methods

This was a single arm, open-label, phase II feasibility study (Australian New Zealand Clinical Trial Registry Number—ACTRN12618001586202). We recruited adults ($\geq$ 18-years-old) with advanced life-limiting illnesses referred to four palliative care services in Sydney, Australia, diagnosed with major depressive disorder from any care setting. Participants received weekly subcutaneous ketamine infusion (0.1–0.4mg/kg) over two hours using individual dose-titration design. Outcomes assessed were feasibility, safety, tolerability and antidepressant activity.

### Results

Out of ninety-nine referrals, ten participants received ketamine and were analysed for responses. Accrual rate was 0.54 participants/month across sites with 50% of treated

PLOS ONE policies on sharing data and materials. However, on further clarification, sharing of data and materials is restricted by the research ethics approval granted for this project (reference number: HREC/18/LPOOL/466), restricting data sharing past the investigator group to reassure participants about the confidentiality of the data. The contact information for the South Western Sydney Local Health District Human Research Ethics Committee: SWSLHD-Ethics@health.nsw. gov.au.

**Funding:** The study was funded by the Translational Cancer Research Network Clinical PhD Scholarship Top-up award (13114323), supported by the Cancer Institute New South Wales, Australia. The funder has no role in the study design, the collection, analysis and interpretation of data, the writing of the report, and the decision to submit the article for publication.

**Competing interests:** CS is the recipient of the Sydney Partnership for Health, Education, Research and Enterprise (SPHERE) Palliative Care Clinical Academic Group Seed Grants. RC receives consulting fees or payments from Cymra Life Sciences Limited, Tilray Australia and New Zealand, AstraZeneca Pty Limited, A. Menarini Australia Pty Ltd. RC also receives payments for expert testimony from Office of the Director of Public Prosecutions NSW. AB assists in running a tertiary referral ketamine service at Black Dog Institute, running the Spravato (intranasal esketamine) early access program, and acting as a site principal investigator on a related quality of life study. CL participated in Janssen Advisory Board, acted as an unpaid consultant for Douglass Pharmaceuticals, and received support from Royal Australian and New Zealand College of Psychiatrists (RANZCP) for presenting at RANZCP Congress. DC is a paid consultant and advisory board member for Helsinn Pharmaceuticals, a paid consultant for Mayne Pharma International Pty Ltd, a paid subcontractor for Nous Group Pty Ltd, a paid board member for Icare Dust Diseases Board, and unpaid consultants for Chris O'Brien Lifehouse and Illawarra Health and Medical Research Institute (IHMRI). DC also receives payment from Mayne Pharma International Pty Ltd for intellectual property. Other authors declare no conflict of interest. This does not alter our adherence to PLOS ONE policies on sharing data and materials. There are no patents, products in development or marketed products associated with this research to declare.

participants achieving $\geq$ 50% reduction in baseline Montgomery-Åsberg Depression Rating Scale, meeting feasibility criteria set *a priori*. There were no clinically relevant harms encountered.

## Conclusions

A future definitive trial exploring the effectiveness of subcutaneous infusion of ketamine for major depressive disorder in the palliative care setting may be feasible by addressing identified study barriers. Individual dose-titration of subcutaneous ketamine infusions over two hours from 0.1mg/kg can be well-tolerated and appears to produce transient antidepressant signals over hours to days.

## Introduction

Individuals with palliative care needs facing advanced cancer and non-malignant life-limiting illnesses often experience a myriad of stressors that predispose them to the development of major depressive disorder (MDD), affecting 10–20% of this population [1–5]. Importantly, MDD can worsen the sense of suffering for individuals, exacerbates co-existing physical symptom distress, and impairs people's ability to engage with their family and friends for important conversations when quality time is critical [6, 7]. It is also associated with suicidal ideation and the desire for hastened death, contributing to psycho-existential distress and poor quality-of-life [2, 6, 8, 9].

Recognising, assessing, and managing MDD can be challenging in the palliative care setting [10–13]. The diagnosis of MDD can be questionable when symptoms of life-limiting illnesses mimic the somatic symptoms of depression [10–12]. The tolerability of interventions might often be restricted by co-existing symptom burdens (e.g., fatigue, breathlessness, cognitive impairment and dysphagia) and organ dysfunction, while interventions lack supportive evidence [14–16]. The progression of the underlying disease and the associated symptom burden may contribute to the development of MDD in the last days to weeks of life, with some individuals developing MDD de novo while others with pre-existing histories of such relapsing [3, 17, 18]. There is a need for tolerable therapeutic options that have rapid-onset actions, particularly towards the end-of-life when typical antidepressants may not provide symptomatic benefits in time [19, 20].

Ketamine, an N-methyl-D-aspartate (NMDA) antagonist used as a dissociative anaesthetic, has been found to exhibit rapid-onset antidepressant activity in otherwise well people with treatment-resistant depression at sub-anaesthetic doses of 0.5mg/kg given over 40 minutes intravenously or as subcutaneous boluses, and is generally well-tolerated [21–23]. It is thought to rapidly elevate extracellular glutamate levels in the brain, increasing alpha-amino-3-hydroxy-5-methyl-4isoxazeolepropionic acid (AMPA) receptor activation and brain-derived neurotrophic factor (BDNF) level in the prefrontal cortex and the hippocampus [24]. Subsequently, there is an increase in synaptogenesis and neural plasticity, postulated to result in the antidepressant effect [24]. The onset of the antidepressant effect can be as rapid as two hours post-infusion of ketamine given parenterally, with its effect lasting up to one week as a single bolus dose or 12 weeks as repeated boluses [21, 23–31]. The response rate has been as high as 70%, with the number needed to treat of three on a meta-analysis [21, 26, 29]. Ketamine, thus, may be a rapidly effective agent to treat MDD in the palliative care setting or a "bridging intervention" while waiting for typical antidepressants to work.

However, in the literature, there are concerns about ketamine causing neurocognitive adverse effects when used in the palliative care setting, even when given at lower dosages [32]. These cast doubt on the tolerability of this conventional ketamine dose in the psychiatry literature when applied to the broader palliative care population. Meanwhile, there is growing evidence to suggest that ketamine's tolerability may be enhanced with antidepressant activity maintained when given as subcutaneous infusions over two hours using a weekly individual dose-titration design, titrating from the ultra-low dosage of 0.1mg/kg [23, 33–37]. Apart from retrospective studies and prospective case reports/series, little is known about ketamine's efficacy and tolerability in people with advanced life-limiting illnesses, particularly in the last days to weeks of life [23, 37, 38].

Prior to conducting a definitive trial to assess ketamine's effectiveness as an antidepressant compared to standard care, the feasibility of conducting such a study needs to be explored, along with its effects and tolerability in this population [37]. This study aims to determine the feasibility, safety, tolerability, acceptability and antidepressant signal/activity to justify and inform a fully powered study of subcutaneous ketamine infusions for MDD in the palliative setting.

## Methods

This study is reported according to Consolidated Standards of Reporting Trials (CONSORT) extension for randomised pilot and feasibility trials Checklist (S1 Checklist) [39]. In-depth details and rationales of the study design and procedures are included in the published protocol paper in BMJOpen (2021) (S1 File) (Australian New Zealand Clinical Trial Registry Number: ACTRN12618001586202) [37]. A summarised methods section is presented here. All participants provided written informed consent.

### Design

This phase II feasibility study used a single-arm, open-label, individual dose-titration design.

### Population

The study inclusion criteria were consented adults ($\geq$ 18-years-old) with advanced life-limiting illnesses known to palliative care services, screened positive with Patient Health Questionnaire-2 (PHQ-2) scores $\geq$ 3 [40, 41], fulfilled the Endicott diagnostic criteria for MDD [37, 42] and assessed to have a Montgomery-Åsberg Depression Rating Scale (MADRS) $\geq$ 16 [43, 44]. Participants were recruited from four palliative care services in Sydney, Australia (St George/Calvary Hospital, St Vincent's Hospital/Sacred Heart, Liverpool Hospital and Braeside Hospital) between July 2019 and October 2021. They included large university teaching hospitals, associated palliative care units and community palliative care services.

The exclusion criteria, in brief, were: Australian-modified Karnofsky Performance scale (AKPS) score = 10; methylphenidate use in the last four weeks; changes to anti-depressant doses in the last two weeks before the commencement of ketamine; ketamine use in the last four weeks; previous significant adverse effect or hypersensitivity to ketamine; concurrent phenobarbitone use; factors of increased risk of intracranial pressure, sympathomimetic response, and intraocular pressure as detailed in the protocol paper [37]; severe hepatic impairment (bilirubin $\geq$ three times upper limit of normal; aspartate aminotransferase [AST] or alanine transaminase [ALT] > five times upper limit of normal); severe renal impairment (creatinine clearance <15ml/min by Cockroft Gault Equation); lifetime history schizophrenia, bipolar and mania; and recent substance misuse.

## Intervention

An individual dose titration with weekly dosing, assimilating the design of Loo et al. (2016), was used to optimise participants' safety and tolerability towards ketamine [23, 37]. Commercially available preparation of ketamine hydrochloride (Ketalar by Pfizer Australia) was administered. Participants received an initial infusion of 0.1mg/kg ketamine given subcutaneously over two hours. Depending on their antidepressant responses defined by a Montgomery-Åsberg Depression Rating Scale (MADRS), further doses could be administered weekly (seven days) for up to four weeks. The dose could be incremented by 0.1mg/kg weekly if improvement from baseline depression score at day 7 was <25% from the previous dose, with a maximum dose of 0.4mg/kg (S1 Fig) [37]. In contrast, if there has been ≥25% improvement of the depression score from baseline, but the participant was not in remission, ketamine was repeated at the same dose as the previous week. If the participant was in remission, no ketamine was administered until relapse occurred, when the last effective and tolerable ketamine dose was repeated at the weekly time points. If there were safety or tolerability concerns, clinicians would assess the participant to consider ceasing the ketamine treatment or repeating ketamine at the previously tolerable dose. *Positive response* was defined by a MADRS baseline score reduction of ≥ 50%, *remission* as MADRS score ≤ 9, and *relapse* as MADRS ≥ 16 after a prior *remission* [23, 45].

For concomitant medications, typical antidepressant uses and changes in doses were permitted, provided that they were done 48 hours apart from the ketamine administration [37]. This was to comply with the ethics committee's requirement to ensure that participants with poor prognoses were not deprived of the potential benefits of typical antidepressants. Given the slow onset of antidepressant effects of typical antidepressants (≥ four weeks) and the rapid on- and off-set actions of ketamine (within days), the antidepressant effects of ketamine and the typical antidepressant were deemed differentiable [19, 21]. Medications that might cause confounding effects on the measured outcome were not allowed, as listed in the exclusion criteria.

## Control

Due to ethical and feasibility concerns for participants, and the primary objective of this study being to explore the feasibility of a future definitive trial, it was deemed that having no control group would be the most appropriate. Subsequently, there was no randomisation process.

## Outcomes

The primary outcome was feasibility, measured as the absolute numbers and proportions of palliative care patients who have consented, were screened for MDD, achieved study eligibility, received ketamine, followed up, and completed the study with reasons for inability to complete any study stages recorded. Feasibility criteria for a future definitive trial were set *a priori* arbitrarily: a recruitment rate of > 0.5 participants per month; and the proportion of treated participants with a positive response (≥ 50% reduction in baseline MADRS score) in symptoms > 10%.

For secondary outcomes: to measure safety, tolerability and acceptability, general non-psychiatric adverse events by vital signs and gradings of National Cancer Institute Common Terminology Criteria for Adverse Events (NCI CTCAE v4.03), and psychiatric adverse events of psychotomimetic and dissociative symptoms by Brief Psychiatric Rating Scale (BPRS) and Clinician Administered Dissociative States Scale (CADSS) were used respectively; and for measuring the antidepressant signal, the Montgomery-Åsberg Depression Rating Scale (MADRS) was used.

Baseline MADRS and toxicity assessments (including AKPS, vital signs, NCI CTCAE, BPRS, and CADSS) were performed immediately in the inpatient or day-hospital setting before ketamine administration (Day 0). Subsequent toxicity assessments were performed every 30 minutes during the 2-hour ketamine infusion and two and four hours after infusion completion. MADRS was reassessed four hours after infusion completion. Participants were further monitored for changes in NCI CTCAE, AKPS and MADRS scores on Day 1, 3, and 7 of Week 1–4 and Day 7 of Week 5–8.

Ketamine was ceased, or participants were withdrawn from the study when the participants, treating physicians, or research clinicians found the study intervention or participation unacceptable for participants at any time point (e.g., participants clinically deteriorated from their underlying medical conditions, preventing ongoing study participation). Investigators reported any Serious Adverse Event, as defined in the protocol, to the Australian national Palliative Care Clinical Studies Collaborative (PaCCSC) Trial Coordinating Unit, who then liaised with the assigned medical monitor for review and involved the Human Research Ethics Committee if required. Given the feasibility nature of this study, a medical monitor rather than a data monitoring committee was used.

## Quality control and data management

The study was supported and overseen by the Scientific Advisory Committee (SAC) and Trial Management Committee (TMC) of PaCCSC. The site research nurses and investigators assessed study outcomes according to the assessment schedule detailed in S1 Table. To maintain data validity and inter-rater consistency, the chief principal investigator provided regular site training and calibration sessions, using calibrated automated devices for toxicity assessments of vital signs. Data confidentiality, accuracy and protocol compliance were monitored by members of TMC and their delegates and audited on an ad-hoc basis. All participants were allocated unique identification numbers. Trial data were collected and entered by the site research nurses into Research Electronic Data Capture (REDCap), a centralised electronic database protected via Secure Sockets Layer encryption [46]. Study records will be maintained for 15 years after study completion in compliance with National Health and Medical Research Council and the Good Clinical Practice guidelines [47, 48].

## Consenting process and research ethics

As MDD screening was not part of routine clinical care, it was part of the ethics requirement to conduct a capacity assessment for potential participants before depression screening with PHQ-2 [49–51]. Research clinicians were trained to use a modified MacArthur Competence Assessment Tool for Clinical Research when assessing capacity to consent using the participant information sheet, exploring the domains of understanding appreciation, reasoning, and expression of choices [49–51] (S2 Table). Only individuals who provided informed consent could proceed with the study. This study, including the consenting procedure, has been approved by South Western Sydney Local Health District Human Research Ethics Committee (reference number: HREC/18/LPOOL/466).

## Sample size and data analysis

A formal conventional power and sample size calculation was not required due to the feasibility focus of this study, as the study results would inform such for the future definitive trial [52, 53]. Nonetheless, for operational considerations, recruitment for up to two years or a sample size of up to 32 was used as the stopping criterion (assuming a conservative effect size of 30% and an one-sided confidence interval of 80% [54, 55]). Data were analysed with descriptive

statistics without inferential statistics or formal hypothesis testing [53]. These were described and summarised with mean and standard deviation for normally distributed data, and medians and interquartile ranges for non-normally distributed data.

## Results

### Feasibility

Ninety-nine referrals were made to the trial– 97 from palliative care services (Inpatient: n = 88; community: n = 4; consult: n = 4; and research: n = 1), one from medical oncology, and one from psychiatry in sites without formal depression screening protocols (Fig 1). The referrals were primarily by physicians (n = 92) and nurses (n = 7). Within a period of ≤1 week from the referral, only 27 individuals proceeded to consent capacity assessment. The reasons (multiple reasons allowed for each individual referred) related to: treating clinicians' decision to exclude potential participants from the study on further assessments (n = 33, 41.8%) (e.g., due to the perceived lack of capacity to consent because of rapid clinical deterioration from underlying medical illnesses, and variations in depression assessment outcomes); meeting study exclusion criteria (n = 22; 27.8%); participant declining (n = 20; 25.3%) (e.g. not wanting medications as depression interventions, perceived study burden or competing priorities); and family declining (n = 4; 5.1%). Twenty-six individuals demonstrated having the capacity to consent, with 20 of those providing consent for the study and proceeding to screening for MDD using PHQ-2. Sixteen individuals screened positive for depression (PHQ-2 score ≥3). Fourteen people met Endicott Criteria for MDD and had clinically significant severity (MADRS ≥ 16). Out of the fourteen, three did not proceed due to meeting exclusion criteria (glaucoma, changes to

**Fig 1. CONSORT flow diagram.** Abbreviations: MADRS–Montgomery-Åsberg Depression Rating Scale; PHQ-2 – Patient Health Questionnaire-2.

**Table 1. Demographics of participants.**

| | N / Median (Interquartile Range [IQR]) |
|---|---|
| Number of participants | 10 |
| Gender (male) | 7 |
| English speaking | 10 |
| Primary palliative diagnoses | |
| Malignant disease | 9 |
| Haematological (myeloma, myelofibrosis) | 2 |
| Solid Tumour* | 7 |
| Gastrointestinal (including colorectal cancer) | 3 |
| Lung (Non-small cell lung cancer) | 3 |
| Breast | 1 |
| Prostate | 1 |
| Stage IV/Metastatic disease | 7 |
| Brain | 2 |
| Lung | 4 |
| Liver | 3 |
| Bone | 5 |
| Peritoneum | 0 |
| Non-malignant disease | 1 |
| COPD–severe | 1 |
| Clinical symptoms of depression | 10 |
| Pre-existing diagnosis of depression | 7 |
| History of treatment resistant depression (failing $\geq$ 2 lines of antidepressants) | 4 |
| Baseline | |
| AKPS | 40 (20) |
| PHQ-2 score | 5.0 (2.8) |
| Endicott Criteria score | 7 (1.5) |
| MADRS score | 32.0 (9.5) |
| BPRS score | 39.5 (4.5) |
| CADSS score | 1.5 (2.5) |
| Respiratory rate (/min) | 18.0 (9.5) |
| Oxygen saturation (SaO2%) | 97.0% (4.0) |
| Systolic blood pressure (mmHg) | 99.0 (18.0) |
| Heart rate (/min) | 97.5 (18.0) |
| Temperature (°C) | 36.4 (0.5) |

*Multiple-option item–i.e., including one participant with metastatic lung and breast cancers.

antidepressant doses in the last two weeks, and deranged liver function test). Eleven people had eligibility confirmed, but one clinically deteriorated from an underlying condition before ketamine commencement. Ten participants received ketamine and were analysed for responses. Seven had pre-existing diagnoses of depression. Four met the criteria for treatment-resistant depression (failing adequate trials of $\geq$ 2-lines of antidepressants as deemed by treating clinicians) (Table 1). Altogether there were 18 episodes of weekly ketamine administrations with the following dose concentration and ranges: 0.1mg/kg (4-9mg)—n = 14; 0.2mg/kg (8-14mg)–n = 2; 0.3mg/kg (21mg)–n = 1; and 0.4mg/kg (28mg)—n = 1. The number of

**Table 2. Participants' completion of various study stages.**

| Participant ID | Completion of data collection at the end of week (Week [number] Day 7) | | | | | | | | Comments/Reasons for withdrawing from study |
|---|---|---|---|---|---|---|---|---|---|
| | 1 | 2 | 3 | 4 | 5 | 6 | 7 | 8 | |
| 33/10/016 | 0 | 0 | 0 | 0 | 0 | 0 | 0 | 0 | Clinical deterioration from underlying disease reaching an exclusion criterion (Tachycardia with fever deemed unlikely due to study intervention) |
| 33/10/017 | 0 | 0 | 0 | 0 | 0 | 0 | 0 | 0 | Clinical deterioration from underlying disease without reaching exclusion criteria (drowsy with possible delirium three days after completing ketamine infusion, deemed unlikely due to study intervention) |
| 33/08/027 | 1 | 0 | 0 | 0 | 0 | 0 | 0 | 0 | Participant died from disease progression of underlying metastatic lung cancer on the sixth day of the study |
| 33/02/007 | 1 | 1 | 0 | 0 | 0 | 0 | 0 | 0 | Condition deteriorated from underlying disease without reaching exclusion criteria (tachycardia on baseline assessment prior to receiving study intervention onset seven days after completion of ketamine infusion) |
| 33/10/009 | 1 | 1 | 0 | 0 | 0 | 0 | 0 | 0 | Logistical burden—no longer wished to be bothered (Self-withdrawn) |
| 33/02/003 | 1 | 1 | 1 | 0 | 0 | 0 | 0 | 0 | Clinical deterioration from underlying disease without reaching exclusion criteria (fatigue from disease progression and unable to engage in interview) |
| 33/10/019 | 1 | 1 | 1 | 1 | 1 | 0 | 0 | 0 | Clinical deterioration with occurrence of an exclusion criterion (atrial flutter prior to receiving the second dose of study intervention, deemed unlikely due to study intervention) |
| 33/08/024 | 1 | 1 | 1 | 1 | 1 | 0 | 0 | 0 | Logistical burden–"had enough" (Self-withdrawn) |
| 33/08/013 | 1 | 1 | 1 | 1 | 1 | 1 | 1 | 1 | Clinical deterioration from underlying disease without reaching exclusion criteria (frailty and severe hearing impairment, impairing study assessments) |
| 33/08/019 | 1 | 1 | 1 | 1 | 1 | 1 | 1 | 1 | "Very happy" for the trial processes involved but felt trial processes were at times burdensome when feeling unwell |

Green: Completion of study collection at end of the week (Week [number] Day 7)

Red: Incompletion of study collection at the end of the week (Week [number] Day 7)

ketamine doses received by the ten participants through the study was: one (n = 6); two (n = 2); and four (n = 2).

The accrual rate, adjusting for the months of trial closure due to COVID-19 related issues, was 0.54 participants per month across all sites. Attrition and completion rates of participants for each study stage are shown in Table 2, with eight participants reaching the end of week 1, four at the end of week 4, and one at the end of week 8 (study completion). Six participants were withdrawn from the study due to clinical deterioration from underlying illnesses rendering further study participation not possible. Among these, two reached withdrawal criteria based on study exclusion criteria. One died due to the underlying disease. Two participants self-withdrew due to the logistical burden. One person completed the study until the end of the follow-up.

The primary reason for attrition was due to clinical deterioration from the underlying life-limiting illnesses and associated complications. No participants withdrew from the study due to unacceptable toxicity or intolerance to study intervention as determined by the participant or site investigator. Participants reported finding trial processes burdensome when feeling clinically unwell from the underlying terminal illness and its complications, contributing to the attrition rate. Frailty and functional decline posed logistical challenges for participants.

## Safety & tolerability

**Psychotomimetic/Dissociative effects.** There was no clinically relevant increase in psychotomimetic (BPRS) or dissociative (CADSS) effects at any time points (0, 2hr, 4hr, 6hr) from baseline.

**Heart rate.** One participant had a gradual increase in heart rate at 1-hour after commencing the infusion at 0.3mg/kg and peaked with an increase over baseline of 51%, reaching 98

beats/min 4-hour post infusion completion. The participant attributed this to the question-naire burden, although study intervention can increase irritability. The heart rate increase for that participant did not recur at the higher dose of 0.4mg/kg in the subsequent week. For the rest of the group, there was no clinically meaningful increase of median change from baseline heart rates at all time points on the intervention days (max +3.1%; +2.0 beats/min).

**Blood pressure.**   There was a dose-response increase in baseline systolic blood pressure with the highest increase of median change from baseline to be 7.9% (8.0mmHg), 12.8% (13.0mmHg) and 16.3% (16.0mmHg) for 0.1mg/kg, 0.2mg/kg and 0.3mg/kg doses respectively at the end of the ketamine infusion. The systolic blood pressure then normalised over the next four hours.

**Other adverse events.**   For expected adverse events known to be associated with ketamine, apart from having a mild increase in the prevalence of grade 2 somnolence (11% - 17%) and grade 1 headache (11% - 17%) no longer than two hours of the infusion duration, there was no significant harm encountered.

Apart from the expected adverse events, there were 86 reported adverse events. Regarding the causality to study intervention, 71 events (82.6%) were deemed "unrelated", 12 events (14.0%) "unlikely", three events (3.5%) "possible" (two events being grade 1 hypertension with systolic blood pressures between 120 and 139), and one being borderline sinus bradycardia. None of these events was serious in nature. There was one serious adverse event for hospitali-sation due to the underlying disease (imaging for possible spinal cord compression in the con-text of myeloma), deemed unrelated to the study intervention.

**Antidepressant effects.**   The total MADRS scores of individual participants over time are illustrated in Fig 2. Out of 18 episodes of ketamine administrations among the 10 participants,

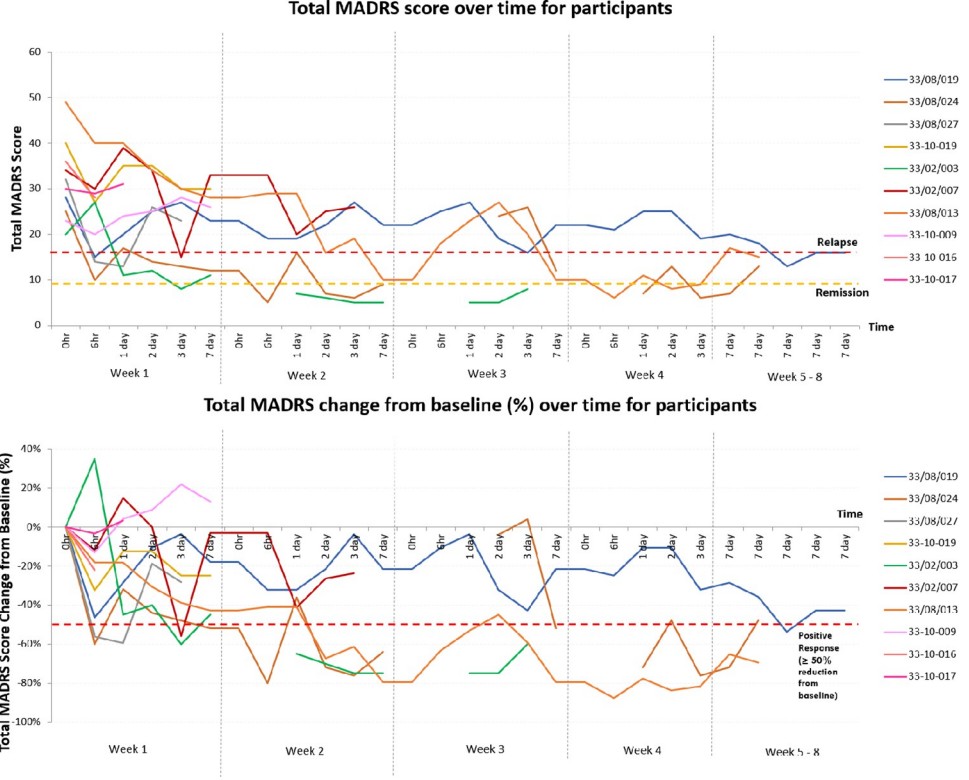

**Fig 2. Total MADRS score changes over time for participants. Abbreviation: MADRS—Montgomery-Asberg Depression Rating Scale.**

there were six occasions of positive responses ($\geq$ 50% reduction in weekly baseline MADRS score) in five participants, all occurring after 0.1mg/kg infusions during week 1 (n = 4) and week 2 (n = 2). Three of the six occasions of positive responses occurred six hours after the infusion commencement (first dose– 0.1mg/kg: n = 2; second dose– 0.1mg/kg: n = 1), two on day 3 (both first dose—0.1mg/kg) and one on day 7 (second dose– 0.1mg/kg). Only two participants received dosages $\geq$ 0.2mg/kg with mixed responses: one participant achieved a positive response and remission with the highest dosage received being 0.2mg/kg; and another participant not achieving a positive response despite dose escalation to 0.4mg/kg.

For the prevalence of positive response ($\geq$ 50% reduction in individuals' baseline MADRS score from study commencement) by participants over the study course, using the total MADRS score, five out of 10 participants (50%) had positive responses during the week of having received ketamine infusion, with two participants having sustained positive response on day 7 after the infusion. All five positive responses occurred with the 0.1mg/kg dosing. As clinical deterioration from underlying life-limiting illnesses can mimic somatic items of MDD, on excluding the somatic MADRS items, there were more participants with positive responses (six of ten). Three of the six participants who developed positive responses during the week had sustained positive responses on day 7 post-infusion. For suicidal ideation (MADRS item 10), nine of ten participants had a $\geq$ 50% reduction in baseline suicidal ideation during the intervention week. Seven of these nine participants had sustained $\geq$ 50% reduction in baseline suicidal ideation on day 7 post-infusion.

Three participants achieved remission (MADRS $\leq$ 9) throughout the study. Out of these, two relapsed (MADRS $\geq$ 16) within seven days of remission.

## Discussion

### Feasibility of the future definitive trial

The study finds that conducting a definitive ketamine trial for MDD in the palliative care setting involving people with extremely short prognoses of days to weeks of life may be feasible (as indicated by the recruitment rate and the positive response rate set *a priori*). Trial recruitment was severely impacted by the intermittent shutdowns of the trial due to COVID-19. Other recruitment barriers were primarily related to clinicians' challenges of screening, assessing and managing depression in the palliative care setting while navigating through participants' rapidly deteriorating medical conditions. This echoes the literature findings of the perceived lack of palliative care psychiatry training by palliative care and psychiatry clinicians, highlighting the need for such at local participating sites before the future definitive study [11, 13]. A better integration between palliative care and psychiatry services locally might be able to improve the necessary palliative care psychiatry skills for clinicians, facilitating routine screening and timely access of mental health interventions to participants when time is critical [11, 13].

### Tolerability & antidepressant activities

Unlike the ketamine burst protocol for cancer pain (100-500mg/day) that was difficult to tolerate for participants [32], this study shows the use of ultra-low dosages of ketamine for depression starting at 0.1mg/kg over two hours is safe and well-tolerated. Additionally, this study demonstrated rapid-onset (even four hours post-infusion) antidepressant activities of ketamine at dosages less than the conventional psychiatric dosing of 0.5mg/kg given via the subcutaneous infusion route [21, 23]. The effect size appears promising, with a 50% positive response rate during the week of intervention using total MADRS scores, and even higher if excluding somatic symptoms of depression. However, in line with the literature, this

antidepressant activity appeared short-lived (hours to days) [25, 56]. Two out of three participants with remission relapsed within one week. To generate a sustained antidepressant activity, a repetitive dosing with an interval more frequent than once a week (e.g., twice or thrice weekly) may be required [31, 57].

## Capacity to consent

Concurring with the literature, this study provides evidence that treating clinicians might act as gatekeepers of potential participants from clinical research [58]. This might be related to their perceptions of how unwell the potential participants were, assuming their lack of capacity to consent, and that participation may cause undue burden to these individuals without meaningful benefits [58]. Interestingly, for those who were allowed to proceed to consent capacity assessment, most potential participants retained the capacity to consent despite their high degree of frailty. Furthermore, while the study assessment tools were found to be burdensome, the actual intervention was well tolerated and produced timely but transient antidepressant activities for the majority of participants. Within the limitation that these participants might have already been selected by the treating clinicians to retain capacity to consent, future clinicians might consider lowering the "gatekeeping" threshold for individuals to participate in research and allowing individuals to have formal capacity assessment as part of the trial assessment by the research team [58]. This may maximise individuals' chances of receiving potentially meaningful benefits from trial participation [58].

## Implications

This study has raised several implications at the clinician, health system, policy and research levels. Ketamine warrants exploration in future definitive studies, evaluating its role in the symptomatic treatment of MDD with or without typical antidepressants when prognoses are extremely short. Screening and assessment of MDD in the palliative care setting using ultra-short questionnaires (e.g. PHQ-2) and Endicott Criteria might be feasible, and may be further investigated [40, 42]. For the feasibility of future trials in this setting, consideration should also be given to non-conventional study designs (e.g., aggregated n-of-1 design or Bayesian response adaptive randomisation) [16, 59, 60]. Given the rapid clinical deterioration of participants, future studies should consider shortening the study duration to improve feasibility. Balance between narrowing the eligibility criteria to maximise detectable treatment response versus broadening the eligibility to enhance recruitment needs to be carefully considered. Future studies can consider the Human Research Ethics Committee's oversight of proxy or early consent as this study did not include severely depressed individuals who could not consent. The collection of brief but regular qualitative data throughout the study course from both participants and carers may help capture objective treatment responses (e.g., social engagement). Lastly, ketamine and esketamine nasal spray may be an alternative option in future studies to enhance study feasibility [61–63].

## Strengths and limitations

To the authors' knowledge, this study is the first prospective study providing key feasibility, safety, tolerability and potential antidepressant activity data for future definitive trial design exploring subcutaneous ketamine for MDD at dosages lower than the conventional psychiatric dose of 0.5mg/kg in a population with extremely short prognoses. It recruited despite multiple trial closures due to COVID-19. It included generous study eligibility criteria to allow for improved generalisability towards the palliative care population with significant co-morbidities and frailty. The individualised dose-titration study design minimised drug toxicity to

participants. The monitoring of toxicity using standardised measures in psychiatry (i.e. BPRS and CADSS) and palliative care research (NCI CTCAE) allowed meaningful comparison of this study with the relevant literature.

There were several limitations of the study for this study. Due to ethical and feasibility concerns, the study was designed as an open-label single-arm design without a placebo or an active comparator arm. This could inflate the true effect size of ketamine through potential assessor bias, Hawthorne effect, and regression to the mean [64–66]. With its primary objective focusing on feasibility, this study was not intended to be powered for inferential statistics, determining the effectiveness of study intervention against the standard-of-care, nor for subgroup analyses to differentiate treatment effects between different dosages of ketamine, first versus subsequent dosing and treatment-resistant status. Building on this feasibility study, future adequately powered studies are needed. There were intrinsic limitations in the assessment of MDD in advanced life-limiting illnesses. Even though Endicott Criteria have been used to reduce the over-identification of cases of MDD, they have not been extensively validated in the palliative care setting, highlighting the need for future research in this field [67, 68].

## Conclusions

A future definitive trial exploring the use of subcutaneous infusion of ketamine for MDD in the palliative care setting may be feasible but challenged by barriers primarily related to the clinicians' challenges in screening, assessing and managing depression in the context of participants' rapidly deteriorating medical conditions. Individual dose-titration of subcutaneous infusions of ultra-low ketamine dosages starting from 0.1mg/kg over two hours can be well-tolerated and produce transient antidepressant activities over hours to days.

## Supporting information

**S1 Checklist. CONSORT checklist.**
(DOC)

**S1 File. BMJ Open study protocol for SKIPMDD.**
(PDF)

**S1 Fig. SKIPMDD study procedure.** Abbreviations: BPRS—Brief Psychiatric Rating Scale; CADSS—Clinician Administered Dissociative States Scale; MADRS—Montgomery-Asberg Depression Rating Scale; PHQ-2—Patient Health Questionnaire-2. *Baseline MADRS score is the MADRS score prior to the last ketamine dose (default) if relapse (MADRS of $\leq 9$) has not occurred. If relapse has occurred, the MADRS score at relapse becomes the baseline.
(DOCX)

**S1 Table. Assessment schedule.** Abbreviations: AKPS—Australia-modified Karnofsky Performance Scale; BPRS—Brief Psychiatric Rating Scale; CADSS—Clinician Administered Dissociative States Scale; ECG–Electrocardiogram; EUC–Electrolyte Urea Creatinine; FBC–Full Blood Counts; LFT–Liver Function Test; MADRS—Montgomery-Asberg Depression Rating Scale; NCI CTCAE—National Cancer Institute Common Terminology Criteria for Adverse Events; PHQ-2—Patient Health Questionnaire-2; TFT–Thyroid Function Test.
(DOCX)

**S2 Table. Consent capacity assessment form.**
(PDF)

## Acknowledgments

We would like to acknowledge the following people for their support: Ms Linda Brown & Belinda Fazekas (Palliative Care Clinical Studies Collaborative [PaCCSC], University of Technology Sydney); PaCCSC research nurses Ms Noula Basides, Ms Frances Bellemore, Ms Robin O'Reilly, Ms Julie Wilcock and Ms Angela Rao; Prof Phillip Good (St Vincent's Private Hospital Brisbane, Kangaroo Point, QLD, Australia); and Dr Amy Chow & Dr Mariclaire Francisco (Braeside Hospital, Prairiewood, NSW).

## Author Contributions

**Conceptualization:** Wei Lee, Caitlin Sheehan, Adam Bayes, Colleen Loo, Brian Draper, Meera R. Agar, David C. Currow.

**Data curation:** Wei Lee, Caitlin Sheehan, Richard Chye, Sungwon Chang, Colleen Loo, Brian Draper, Meera R. Agar, David C. Currow.

**Formal analysis:** Wei Lee, Sungwon Chang, Colleen Loo, Brian Draper, Meera R. Agar, David C. Currow.

**Funding acquisition:** Wei Lee, Brian Draper, Meera R. Agar, David C. Currow.

**Investigation:** Wei Lee, Richard Chye, Adam Bayes, Colleen Loo, Brian Draper, Meera R. Agar, David C. Currow.

**Methodology:** Wei Lee, Caitlin Sheehan, Richard Chye, Sungwon Chang, Adam Bayes, Colleen Loo, Brian Draper, Meera R. Agar, David C. Currow.

**Project administration:** Wei Lee, Caitlin Sheehan, Richard Chye, Adam Bayes, Meera R. Agar, David C. Currow.

**Resources:** Wei Lee, Caitlin Sheehan, Richard Chye, Sungwon Chang, Adam Bayes, Colleen Loo, Brian Draper, Meera R. Agar, David C. Currow.

**Software:** Sungwon Chang.

**Supervision:** Colleen Loo, Brian Draper, Meera R. Agar, David C. Currow.

**Validation:** Adam Bayes, Colleen Loo, Brian Draper.

**Visualization:** Wei Lee.

**Writing – original draft:** Wei Lee.

**Writing – review & editing:** Wei Lee, Caitlin Sheehan, Richard Chye, Sungwon Chang, Adam Bayes, Colleen Loo, Brian Draper, Meera R. Agar, David C. Currow.

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
