## [Decision Letter · Decision Letter 0]

16 May 2023

PONE-D-23-04868Subcutaneous Ketamine Infusion in Palliative Patients for Major Depressive Disorder (SKIPMDD)PLOS ONE

Dear Dr. Lee,

Thank you for submitting your manuscript to PLOS ONE. After careful consideration, we feel that it has merit but does not fully meet PLOS ONE’s publication criteria as it currently stands. Therefore, we invite you to submit a revised version of the manuscript that addresses the points raised during the review process.

We look forward to receiving your revised manuscript.

Kind regards,

Dickens Akena, Ph.D

Academic Editor

PLOS ONE

2. Please describe in your methods section how capacity to provide consent was determined for the participants in this study. Please also state whether your ethics committee or IRB approved this consent procedure. If you did not assess capacity to consent please briefly outline why this was not necessary in this case.

“CS is the recipient of the Sydney Partnership for Health, Education, Research and Enterprise (SPHERE) Palliative Care Clinical Academic Group Seed Grants. RC receives consulting fees or payments from Cymra Life Sciences Limited, Tilray Australia and New Zealand, AstraZeneca Pty Limited, A. Menarini Australia Pty Ltd. RC also receives payments for expert testimony from Office of the Director of Public Prosecutions NSW. AB assists in running a tertiary referral ketamine service at Black Dog Institute, running the Spravato (intranasal esketamine) early access program, and acting as a site principal investigator on a related quality of life study. CL participated in Janssen Advisory Board, acted as an unpaid consultant for Douglass Pharmaceuticals, and received support from Royal Australian and New Zealand College of Psychiatrists (RANZCP) for presenting at RANZCP Congress. DC is a paid consultant and advisory board member for Helsinn Pharmaceuticals, a paid consultant for Mayne Pharma International Pty Ltd, a paid subcontractor for Nous Group Pty Ltd, a paid board member for Icare Dust Diseases Board, and unpaid consultants for Chris O’Brien Lifehouse and Illawarra Health and Medical Research Institute (IHMRI). DC also receives payment from Mayne Pharma International Pty Ltd for intellectual property. Other authors declare no conflict of interest.”

Additional Editor Comments (if provided):

We have received the comments from the authors. Please ensure that you respond to each and every one of the comments raised by the reviewers

Reviewers' comments:

Reviewer's Responses to Questions

**Comments to the Author**

1. Is the manuscript technically sound, and do the data support the conclusions?

Reviewer #1: Partly

Reviewer #2: Yes

Reviewer #3: Partly

Reviewer #4: Partly

2. Has the statistical analysis been performed appropriately and rigorously? 

Reviewer #1: Yes

Reviewer #2: I Don't Know

Reviewer #3: Yes

Reviewer #4: No

3. Have the authors made all data underlying the findings in their manuscript fully available?

Reviewer #1: No

Reviewer #2: Yes

Reviewer #3: Yes

Reviewer #4: Yes

4. Is the manuscript presented in an intelligible fashion and written in standard English?

Reviewer #1: Yes

Reviewer #2: Yes

Reviewer #3: Yes

Reviewer #4: Yes

5. Review Comments to the Author

Reviewer #1: Ketamine research for depression is a topic of relevance and general interest. This study adds to current knowledge by examining this topic in people with advanced life-limiting illnesses. There is a coherent case made of why the research question is important, due to a lack of research on a possible therapeutic option for a specific population. The paper was overall well written and the authors performed careful and thorough processing. However, a few points can be further revised or explained.

1) Introduction can be expanded on with more details.

2) Authors’ should discuss and describe de novo/relapse of MDD more. Also, there is mention of suicide ideation in the result, but not in the introduction.

3) Who does the palliative care population consist of (i.e. life-limiting illnesses)? There is mention of cancer pain literature in the introduction, what else is included in this population and the sample?

4) The design can be expanded on with more details.

5) Is it possible to include the exclusion criteria in the population section of the methods? Rather than just referencing.

6) Why wasn’t a control used? As well as conventional power and sample size calculations not required? Can these be explained in detail?

7) Can the assessment schedule be explained in the outcomes section of the methods? Rather than just referencing.

8) Can authors explain why they used the measures they did? As well as explain what the procedure was for this part of the study.

9) Can the authors discuss the relapse soon after remission in more detail? What are the implications of this?

10) In the implications, there is mention of the eligibility criteria. Can you expand on what this study’s criteria included in detail?

11) The limitations can be expanded on, by including rationale for your decisions and possible solutions for future studies.

Reviewer #2: Important note: This review pertains only to ‘statistical aspects’ of the study and so ‘clinical aspects’ [like medical importance, relevance of the study, ‘clinical significance and implication(s)’ of the whole study, etc.] are to be evaluated [should be assessed] separately/independently. Further please note that any ‘statistical review’ is generally done under the assumption that (such) study specific methodological [as well as execution] issues are perfectly taken care of by the investigator(s). This review is not an exception to that and so does not cover clinical aspects {however, seldom comments are made only if those issues are intimately / scientifically related & intermingle with ‘statistical aspects’ of the study}. Agreed that ‘statistical methods’ are used as just tools here, however, they are vital part of methodology [and so should be given due importance]. I look at the manuscript in/with statistical view point, other reviewer(s) look(s) at it with different angle so that in totality the review is very comprehensive. However, there should be efforts from authors side to improve (may be by taking clues from reviewer’s comments). Therefore, please do not limit the revision only (with respect) to comments made here.

COMMENTS: Although this manuscript is well drafted [and the study is excellent with respect to most of the aspects], I have few observations/concerns (different opinion) which are given below:

Since according to ‘Methods’ section in ‘Abstract’, “This was a single arm, open-label, phase II feasibility study” a mention of this may appear in title {presently it gives the impression that it’s a full-fledged phase-III trial}. Very good it is clearly stated in the ‘Abstract’ itself, such a mention in title is desirable, I guess.

I am sure, authors are well aware of limitations of ‘single arm/group’ study, however, I request them to read a note pasted from one famous standard textbook on ‘Medical Research Methodology’:

It is very essential to keep the limitations in mind while interpreting results from any ‘single arm/group’ study. A classical/ideal clinical trial/study needs/requires a concurrently {but similarly} handled/treated appropriately selected/chosen control/comparison parallel group/arm. Note further that “Inferential statistics (i.e., hypothesis testing + estimation of CI) is built on the population model [which means the underlying assumption is that there is/are population(s) and we are dealing with random sample(s) drawn from that/those population(s)]. Although in clinical trial (involving at least two groups) we do not really deal with random samples (generally a non-probabilistic convenience sampling), ‘allocation’ to treatment groups is ‘randomly’ done which enable us to evoke the population model and we can use inferential statistics safely. But when there is only one group (so that there is no question of random allocation), with ‘non-random’ selection, it may be questionable to use inferential statistics even if you have two measurement sets as ‘pre-post’ or many repeated measurements or use ‘internal grouping for comparison.”

Except these minor points, the article is acceptable. Since ‘when the study is ‘pilot’ in nature, many things are ignored [loosely looked at / evaluated] {example: sample size is not a big issue, other methodology issues need not be looked at very rigorously, etc.’ However, mind you that as pointed out in ‘important note’ above “This review pertains only to ‘statistical aspects’ of the study and so ‘clinical aspects’ should be assessed separately/independently. ‘Minor Revision’ is recommended.

Reviewer #3: Given that the main outcome was feasibility acceptability and tolerability, the findings are relevant.

However, there are major limitations in the population selection and the definition of the secondary outcome of remission. These limitations are not due to the authors failure but rather due to the difficulty in assessing for depression and remission in terminally ill patients. These limitations must be clearly highlighted in the discussion section since you plan for future effectiveness studies.

Other comments

Please add acceptability and tolerability as primary outcomes since you have a lot of data on them.

What informed the decision to dose over 2 hours, yet most guidelines prefer rapid infusion or even bolus?

“If the participant was in remission, no ketamine was administered until relapse occurred, when the last effective and tolerable ketamine dose was repeated” Remission has been defined as two months symptom free which defers from your operational definition. Also, your references 19 and 35 that explain the remission and relapse criteria for this study do not give evidence for this.

Although the main outcomes were feasibility acceptability and tolerability, the difficulty in diagnosing depression in terminally ill patients is a major limitation to this and future studies. Specifically, the Endicott criteria has not been well validated.

Reviewer #4: 1) Sample size calculation is not provided. The final number of 10 participants is not sufficient to support the conclusions of the study.

2) Was randomisation done? if not how was the risk of bias minimised in the study?

3) How was data management achieved and quality control

4) How was data analysis performed?

5) Was the clinical trial registered? if so provide the registration number and if not give reason as to why the trial is not registered

6) How was feasibility and Tolerability measured

7) What safety indices were targeted in this study and how were they defined and measured?

8) What were the effect measures of the study and how was the analysis done?

9) What dose of Ketamine showed optimal antidepressant activity?

10) Background: Include the basis of antidepressant activity of ketamine

11) Include an operational definition of feasibility, tolerability and acceptability needs to be provided. What do these terms mean

12) How was feasibility, tolerability and acceptability measured in this study

13) Who was bias minimised in this study?

14) Who collected the data and how was quality ensured during the data collection

15) What does weekly dosing mean? clearly state the dosing intervals used in the study and the justification

16) How many doses of ketamine did the participants receive?

17) What was the source of the ketamine used in this study (manufacturer, certificate of analysis)

18) What was the stopping criteria used in this trial

19) How was patient safety ensured?

20) Dis the trial have SAC?

21) The methods section lacks details of how the different study outcomes were measured

22) The tools used in measurement of study outcomes eg, MADRS, CADSS, AKPS, BPR, NCI, CTCAE etc needs to be described for clarity

23) If conventional power calculation was not required, how did you arrive at the 10 participants. Additionally, the design of the study should be adjusted to match the number of participants and the methods used in the study

24) Provide details of the different types of cancers that were included in the study

25) How was participant screening for inclusion into the study done, provide the criteria followed

26) Were baseline assessment of the potential study participants does? if so provide details and if not why not

27) How was BP measured and how was quality control ensured

28) On average, how many times was each patients administered Ketamine? and was this included in the interpretation of the results?

29) Were the participants taking other medicines? if so provide details of these medicines and how did these medicine affect interpretation of the study findings?

30) What is clinical deterioration and how was it measured in this study?

31) The result....'Three participants achieved remission......' on pg7 is not clear. What do you means and how was this measured?

32) The clinical characteristics of the study participants lack detail and thus not clear

33) Results on feasibility. There is discussion section, however there are no results on the result section. Additionally, description of the methods that was used is not provided

34) Discussion is provided on tolerability but how this was measured and the results are not provided including how data was analysed

6. PLOS authors have the option to publish the peer review history of their article (what does this mean?). If published, this will include your full peer review and any attached files.

Reviewer #1: No

Reviewer #2: No

Reviewer #3: **Yes: **Dr. Emmanuel Kiiza Mwesiga

Reviewer #4: **Yes: **Dr. Moses Ocan

---

## [Author Response · Author response to Decision Letter 0]

19 Jul 2023

A tabulated response to reviewers' comments have been attached to allow for easier reading.

---

## [Editor Report · Decision Letter 1]

18 Aug 2023

Subcutaneous Ketamine Infusion in Palliative Patients for Major Depressive Disorder (SKIPMDD) - Phase II Single-arm Open-label Feasibility Study

PONE-D-23-04868R1

Dear Dr. Lee

We’re pleased to inform you that your manuscript has been judged scientifically suitable for publication and will be formally accepted for publication once it meets all outstanding technical requirements.

Kind regards,

Dickens Akena, Ph.D

Academic Editor

PLOS ONE
---

## [Editor Report · Acceptance letter]

29 Aug 2023

PONE-D-23-04868R1 

Subcutaneous Ketamine Infusion in Palliative Patients for Major Depressive Disorder (SKIPMDD) - Phase II Single-arm Open-label Feasibility Study 

Dear Dr. Lee:

I'm pleased to inform you that your manuscript has been deemed suitable for publication in PLOS ONE. Congratulations! Your manuscript is now with our production department. 

Kind regards, 

on behalf of

Dr. Dickens Akena 

Academic Editor

PLOS ONE